# Development of the Mental Number Line Representation of Numbers 0–10 and Its Relationship to Mental Arithmetic

**DOI:** 10.3390/brainsci12030335

**Published:** 2022-03-01

**Authors:** Anat Feldman, Andrea Berger

**Affiliations:** Department of Psychology and Zlotowski Center for Neuroscience, Ben-Gurion University of the Negev, Beersheba 84105, Israel; andrea@bgu.ac.il

**Keywords:** child development, mental number line, arithmetic abilities, anchor points

## Abstract

The internal representation of numbers on the mental number line (MNL) was demonstrated by performing the computerized version of the number-to-position (CNP) task on a touchscreen while restricting response time. We found that the estimation pattern is best fit by a sigmoid function, further denoted as the “sigmoidal model”. Two developmental leaps occurring during elementary school were recognized: (1) the division of the number line into two segments and (2) consistent use of different anchor points on the number line—the left endpoint in first grade, the right endpoint in second grade, and finally the midpoint in third grade. Additionally, when examining the differences between the breakpoints, we found that first graders demonstrated a breakpoint close to 6, which linearly decreased over the years until stabilizing close to 5. The relation between the ability to place individual numbers on a number line and performance of mental arithmetic showed that the consistent use of anchor points correlated significantly with faster responses in mental arithmetic.

## 1. Introduction

Many studies investigating mental number representation assume that numbers and quantities are spatially organized along a mental number line (MNL; e.g., [1,2,3]). However, the exact developmental process of the MNL is still under debate. In their pioneering work, Siegler and Opfer [4] attempted to understand this developmental process by using a number-to-position (NP) task with participants of different ages. In this task, in each trial, a number was presented, and the participant was asked to mark the corresponding position on a bounded number line (on paper/screen), which was labeled with numbers only at its endpoints. They found that accuracy of the number position changed with numerical range and participant age. 

For example, it was shown that second-grade children unevenly placed numbers on a 0–1000 number line as their placement pattern was best fit by a logarithmic rather than by a linear function. The same age group placed numbers more linearly (i.e., resulting in a constant distance between consecutive numbers) on a 0–100 number line. In contrast, adults produced a linear distribution when placing numbers on a 0–1000 number line [4]. This developmental change is known as the logarithmic-to-linear representational shift, and it was replicated in a number of subsequent studies [5,6,7,8]. 

Despite such robust evidence for the logarithmic-to-linear representational shift, there are studies that characterized the pattern of estimated number positions on a number line in a different way. Ebersbach et al. [9] found that children divided the number line into two segments and proposed a “segmented model” when examining number placement on a 1–100 number line. In particular, they argued that children divide the number line into two segments of varying range, depending on the child’s age and familiarity with the numbers, with each segment best described by a distinct linear function. 

Moreover, the slope of the first linear function fitting the numbers in the more familiar range is always steeper compared to the slope of the second linear function fitting the numbers in the less familiar range as demonstrated in Figure 1. According to Ebersbach et al. [9], the two ranges are separated by the change points whose values vary between different age groups. Namely, they reported that the change point of kindergartens was 20, first graders was 25.04, and second graders was 37.53. 

Alternative ways for looking at the pattern of estimated number positions on a number line can be found in the studies of Moeller et al. [10] and Barth and Paladino [11]. The first group examined first graders’ estimation abilities on a 0–100 number line and showed that the participants tended to divide the number line by a breakpoint close to 10. The second group of researchers appealed to the proportion judgment model [12,13], which posits that participants estimate the magnitude of a part relative to the whole. According to this model derived from Stevens’ power Law [14,15,16], the relationship between the estimated magnitude of a stimulus and its actual magnitude is determined by a power function, *y* = *ax^β^*. 

As the number line in a typical NP task is bounded, the estimated position of each number relates to the entire range of the numbers. In other words, it is assumed that participants use the numbers on the edges as anchor points, which helps them to determine the proper position of each number on the given number line. As a result, the estimated number positions fit an s-shaped or an inverse s-shaped curve—both constituting a one-cycle model, as shown in Figure 2A. The one-cycle model can be further extended to a two-cycle model once participants begin to use the midpoint of the number line as an additional anchor point. In this case, the resulting pattern of the estimated number positions would look like a concatenation of two s-shaped curves: one between the left edge and the midpoint and another between the midpoint and the right edge (see a two-cycle model shown in Figure 2B).

Studies investigating developmental differences according to the proportion judgment model within the 0–100 number range [11,17,18,19,20] showed that children tend to overestimate number positions between the left edge (0) and the midpoint (50) and underestimate the number positions between the midpoint (50) and the right edge (100). In addition, studies have shown that the use of the midpoint as an anchor point is established at the beginning of third grade, thus yielding a more accurate number position within the given range [19,21,22].

Based on the above findings, it appears that both the two-linear and the proportion judgment models divide the number line into smaller segments, which helps participants place a given number more accurately. This number line division has also been documented in the literature by showing that children and adults subjectively divided the number line into “landmarks” in an attempt to place numbers more accurately along the number line [4,23]. 

However, the drawback of all the above studies was that they used the same model to fit the placement of both one-digit and two-digit numbers within various number ranges (e.g., 0–20 or 0–100) while ignoring the differences between unit-decade encoding processes inherent in one-digit and two-digit numbers, as stressed in several studies [10,24,25]. As follows from the above literature survey, the determination of fundamental mechanisms governing the developmental process of the MNL is still the focus of active academic research. 

Despite a general agreement regarding the important role of anchor points in assisting children to determine the correct position of a specific number on the number line, the final word in relating the number and position of anchor points to a given range of numbers and to a child’s age has still not been decided. Surprisingly, no systematic studies have been conducted to investigate the developmental process of the MNL with respect to the placement of solely one-digit numbers. This fact can apparently be explained by an assumption of excessive simplicity for children of all ages when performing the NP task with one-digit numbers. 

However, performing the NP task with single-digit numbers has two noticeable advantages: first, it minimizes uncertainties related to distinct mechanisms responsible for processing multi-digit numbers and second, first through six graders are all familiar with this range of numbers, which means that changes observed in the NP task between children can be strictly related to their maturity processes rather than just to exposure to the specific range of numbers. For both reasons, the NP task performed using a one-digit range of numbers can also constitute an excellent platform for its further correlation with arithmetic achievements of children of all ages. 

In fact, because the ability to place numbers on the number line is based on spatial numerical association (SNA) and numerical cardinalities [26,27], researchers have found a significant correlation between performance in number-line estimation tasks and arithmetic achievements [18,28,29,30,31,32,33,34,35,36,37]. All the above-mentioned studies examining the relationship between performance in the number line task and mathematical competence have shown evidence of a correlation between a child’s percentage of absolute accuracy on the number line estimation task and different arithmetic tasks. 

However, a limitation of these studies is that they examined performance only in terms of the mean estimation of all numbers, ignoring the differences between the placements of the individual numbers. As such, there is no last word on critical factors correlating between performance in number-line estimation tasks and arithmetic achievements.

The objectives of our study were twofold. First, we focused on the representation of numbers within the 0–10 number range and proposed a new approach that encompassed properties inherent to both the two-linear and the proportion judgment models. A key idea was that in the process of estimation of placing numbers on the 0–10 number line, children would divide the number line into two segments, and the developmental differences between the groups would be characterized by the flexible breakpoint. 

In this context, we hypothesized that the sigmoidal function, incorporating the features of both the two-linear and the proportion judgment models and taking into account the existence of flexible breakpoint (see Figure 3), would best characterize the number representation on the number line. Second, we examined the correlation between the ability to place individual numbers on a physical number line (focusing on the breakpoint and the numbers on the edges that serve as anchor points) and the performance of mental arithmetic using the arithmetic facts (AF) task (a true/false verification model) with numbers up to 10.

Both objectives were achieved by performing three experiments, including computerized and “classic” paper and pencil number-to-position (NP) tasks as well as the AF task, including a discrepancy/split factor constituting the difference between “close” and “distant” wrong answers [38] based on an estimation strategy [39,40]. Based on the idea that an estimation strategy is dominant both in the NP and the AF tasks, we hypothesized that we would find a correlation between estimation accuracy in each of the NP tasks and RT in the AF task.

## 2. Experiment 1

### 2.1. Method

#### 2.1.1. Participants

A total of 181 right-handed (to avoid any motor differences in the digital task that included finger trajectories, we involved only right-handed children in our study) children from first through sixth grades were recruited for this study from elementary schools within the neighborhood surrounding the university campus (see Table 1 for the participant demographics).

The procedure was approved by the Israeli Ministry of Education. For all child participants, written informed parental consent was obtained. The children were predominantly from middle to upper class families.

#### 2.1.2. Materials and Procedure

The participants met one-on-one with the experimenter and performed the computerized NP (CNP) task in a quiet room. Each participant was presented with a touchscreen displaying a bounded number line that remained at the top part of the screen throughout the experiment. The line had marked numbers only at the endpoints (0 and 10). To start a trial, the participant touched the starting point (a gray rectangle at the middle bottom of the screen), which caused a fixation cross to appear above the middle of the number line. The participant then moved his/her finger across the screen toward the number line. 

The fixation cross was replaced by the target number when the finger reached 10 px above the starting point, and the participant continued moving toward the estimated corresponding position on the 0–10 number line. A feedback arrow showed the point where the participant estimated the position of the presented target number (see Figure 4). Each number between 0 and 10 randomly appeared five times as a target number above the number line. 

The trial was considered invalid if the finger moved backward, was lifted from the screen in midtrial, or moved too slowly (more than 2500 ms from appearance of the target number until the finger reached the number line) or too quickly (less than 800 ms to cross 30% of the vertical distance). Invalid trials were not analyzed, and the target number appeared later again in the experiment. No feedback was given except for general encouragement. The experimenter said, “Here on the touchscreen you see a number line with two numbers on its ends, 0 and 10. You are asked to touch the gray square, then the number will be presented above the middle of the number line, without lifting your finger from the screen, you need to drag your finger to the place you think the number should be”.

The CNP task was preceded by two training tasks:A task aimed to ensure a minimum level of motor control for the participants. The task was administered like the CNP task, except that each stimulus was not a number but a downward-pointing arrow whose tip touched a random position on the number line. The participants dragged their finger to the location of the arrow. There were 20 trials; no feedback was received.Ten training trials in the CNP task were presented to ensure that the participant clearly understood the task. The data set of these trials was not included in the analysis.

The internal reliability of these conditions was estimated by using the Spearman–Brown corrected split-half correlations [41], separating all endpoint scores of each target number into two sets of trials: the odd and the even trials. Then, we calculated the asymmetry score for each set and the Pearson product moment correlation between the two resulting sets of asymmetry scores; the correlation was *r* = 0.95.

### 2.2. Results

#### 2.2.1. The Estimated Position Pattern Analysis

It is noteworthy that all participants successfully completed the above training parts of the task. The accuracy in the arrow task was very high; therefore, we did not include this part in the results section.

First, we examined the mean estimated position of each target number. As the endpoints (0, 10) were presented on the screen, the analysis included only data of the target numbers 1–9 for each grade (see Figure 5).

To characterize the pattern for the estimated positions on the number line, for each participant, we computed the *R*^2^ values of the curve that fit the estimations according to the proportion judgment model and the sigmoidal model. To calculate the sigmoidal function that fit the estimated positions, we used the Levenberg–Marquardt algorithm: $y=\mathrm{Bottom}+\frac{\left(\mathrm{Top}-\mathrm{Bottom}\right)}{\left[1+\exp\frac{\left(\mathrm{breakpoint}-\mathrm{estimated}\;\mathrm{position}\right)}{\mathrm{slope}}\right]}$. The breakpoint (discrete values from 1 to 9) of the sigmoidal function corresponding to each participant was computed using a least square fit, based on maximal individual *R*^2^ values (see Table 2).

As shown in Table 2, there were no significant differences between the two-linear model fit and the sigmoidal model fit in first graders; at this age, it appears that the two-linear model was the best fit followed by the proportion judgment model. However, the sigmoidal model best fit the pattern of the estimated number position for all other grades.

A general consensus is that models with more free parameters typically fit curves better than models with less free parameters. In the current study, the proportional model used one free parameter, the sigmoidal model used two free parameters, and the bilinear model used three free parameters. For a fair comparison between models, the fitting adequacy must be corrected to the number of free parameters [42]. To adjust for this, the Akaike Information Criterion (AIC; [42]), which determines the relative information value of the model using the maximum likelihood estimate and the number of parameters (independent variables) in the model, was used in the current study. 

The formula for AIC is: $\mathrm{AIC}=2k-2ln\left(\hat{L}\right)$, where k is the number of independent variables used and $\hat{L}$ is the log-likelihood estimate. The AIC has been successfully used for number line estimation tasks (e.g., [43,44]). To identify the model that fit most participants, we assessed the AIC of each model for each participant and then averaged the obtained results over all the participants. The averaged AIC results with respect to the participants’ grade are summarized in the last column of Table 2. It can be clearly seen that, on average, the sigmoidal model constituted the best fit for the data acquired in all age groups compared to other models.

To examine the breakpoint differences between grades, we conducted a one-way ANOVA (analysis of variance) with the breakpoint as the dependent factor and grade as the independent factor with six levels (first through sixth grades). The results revealed significant differences between grades, *F*(5, 175) = 21.59, *p* < 0.001, $\eta_{p}^{2}$ = 0.38; the mean breakpoint of first grade was 6, which linearly decreased from second to sixth grade (5.6, 5.34, 5.28, 5.15, and 5.16, respectively), *F*(1, 148) = 94.37, *p* < 0.001, $\eta_{p}^{2}$ = 0.35. 

The correlation analysis between the breakpoint and the overall accuracy in the CNP task indicated significant negative regression, *R*^2^ = −0.71, *t*(1, 179) = 33.94, *p* < 0.001. For the trend analysis of breakpoint differences between grades, Levene’s homogeneity test was performed on the data. A significance of *p* < 0.05 demonstrated that variances were not homogenous; thus, because the data had unequal variances, a Games–Howell multiple comparisons test was performed. 

The results of the one-way ANOVA showed significant differences, *F*(5, 180) = 21.59, *p* < 0.001, $\eta_{p}^{2}$ = 0.37. To sum up, the trend analysis of breakpoint differences between grades indicated significant changes between first graders and older groups and between second graders and older groups; no changes were observed between grades beginning with third grade. 

Next, we examined the bias of each target number as the deviation between the estimated position of each target number and its correct position (see Figure 6). A negative bias meant underestimation of number positions, and a positive bias meant overestimation of number positions. A two-way mixed ANOVA, with bias as the dependent measure, grade (1–6) as a between subjects factor and target number (1–9) as a within subjects factor, revealed a significant main effect for target number, *F*(8, 1400) = 77.13, *p* < 0.0001, $\eta_{p}^{2}$ = 0.31, grade, *F*(5, 175) = 16.48, *p* < 0.0001, $\eta_{p}^{2}$ = 0.32, and the interaction between target number and grade, *F*(40, 1400) = 6.45, *p* < 0.0001, $\eta_{p}^{2}$ = 0.16. 

Further analyses showed a significant difference between first and second grade, *F*(1, 175) = 4.9, *p* < 0.05, $\eta_{p}^{2}$ = 0.02, and between second and third grade, *F*(1, 175) = 7.7, *p* < 0.01, $\eta_{p}^{2}$ = 0.04. First graders underestimated the position of all numbers except the number 1. Second and third graders underestimated the position of numbers 1–6, and fourth through sixth graders underestimated the position of numbers 1–5.

#### 2.2.2. The Estimated Position Accuracy Analysis

The percentage of absolute error of estimation (PAE) of the estimated position of each target was calculated based on the deviation from the correct target position as a percentage of the number line length [4]: $1-\frac{\left|\mathrm{Estimated}\;\mathrm{position}\;-\;\mathrm{correct}\;\mathrm{target}\;\mathrm{location}\right|}{\;10\;}$, see Figure 7. 

The PAE was low for all grades: 12% in first grade, 10% in second grade, 8% in third grade, 7% in fourth grade, and 5% in fifth and sixth grades. To test the effect of grade, we conducted a two-way mixed ANOVA, with PAE as the dependent measure and with grade (1–6) as a between subjects factor and target number (1–9) as a within subjects factor. Consistent with the previous analyses, there was a significant main effect for target number, *F*(8, 1400) = 39.2, *p* < 0.0001, $\eta_{p}^{2}$ = 0.18, grade, *F*(5, 175) = 19.7, *p* < 0.0001, $\eta_{p}^{2}$ = 0.36, and the interaction between target number and grade, *F*(40, 1400) = 5.6, *p* < 0.0001, $\eta_{p}^{2}$ = 0.14. Further analyses showed there were two significant accuracy leaps. 

One leap occurred between first grade and second grade, *F*(1, 175) = 6.68, *p* < 0.05, $\eta_{p}^{2}$ = 0.01, stemming mostly from the accuracy differences for the target number 5, *F*(1, 175) = 54.42, *p* < 0.001, $\eta_{p}^{2}$ = 0.23, and a weaker accuracy difference for target numbers 2, 3, and 6, *F*(1, 175) = 7.9, *p* < 0.01, $\eta_{p}^{2}$ = 0.04; *F*(1, 175) = 4.45, *p* < 0.05, $\eta_{p}^{2}$ = 0.02; and *F*(1, 175) = 11.71, *p* < 0.001, $\eta_{p}^{2}$ = 0.06, respectively. A second leap occurred between second and third grades, *F*(1, 175) = 6.69, *p* < 0.05, $\eta_{p}^{2}$ = 0.01, and stemmed from accuracy differences for target number 4, *F*(1, 175) = 4.4, *p* < 0.05, $\eta_{p}^{2}$ = 0.02, target number 5, *F*(1, 175) = 6.9, *p* < 0.001, $\eta_{p}^{2}$ = 0.04, and target number 6, *F*(1, 175) = 3.98, *p* < 0.05, $\eta_{p}^{2}$ = 0.02. No significant differences were found between other consecutive pairs of grades from third through sixth grades.

The following analyses examined whether the endpoints (0, 10) of the number line and midpoint (5) served as anchor points, helping children better estimate where to place numbers that are close to these anchors (see Figure 7). A quadratic contrast (including the numbers 1–9) and a double-quadratic contrast (one between numbers 1 and 5, the other between numbers 5 and 9) were conducted for each grade. The results revealed a significant quadratic contrast, confirming that first graders tended to better estimate number positions close to the endpoints than other numbers on the number line, *F*(1, 27) = 26.88, *p* < 0.001, $\eta_{p}^{2}$ = 0.5. 

Moreover, the double-quadratic contrast showed that the second through sixth graders tended to significantly better estimate number positions close to the endpoints and the midpoint: second grade, *F*(1, 26) = 30.59, *p* < 0.001, $\eta_{p}^{2}$ = 0.54; third grade, *F*(1, 30) = 55.18, *p* < 0.001, $\eta_{p}^{2}$ = 0.65; fourth grade *F*(1, 28) = 31.55, *p* < 0.001, $\eta_{p}^{2}$ = 0.53; fifth grade, *F*(1, 30) = 98.91, *p* < 0.001, $\eta_{p}^{2}$ = 0.77; and sixth grade, *F*(1, 34) = 55.63, *p* < 0.001, $\eta_{p}^{2}$ = 0.62. 

### 2.3. Discussion

In order to demonstrate the internal representation of numbers on the MNL, we used the computerized version of the number-to-position (CNP) task, which restricts response time and is performed on a touchscreen. We found that the estimation pattern was best fit by the sigmoidal function, which we termed the “sigmoidal model.” This model has the advantage of combining both the proportion judgment model [11] and the two-linear model with a flexible breakpoint [9]. 

Similar to the proportion-estimation model, the sigmoidal model elucidated the use of anchor points during the estimation process. As in the two-linear model, it demonstrated that participants tended to divide the number line by using different breakpoints. We found that two developmental leaps occurred during elementary school. Both developmental leaps were expressed in the division of the number line into two segments and consistent use of different anchor points on the number line: the left endpoint in first grade, the right endpoint in second grade, and the midpoint in third grade. 

The next experiment included children from second and third grades, who (according to the first experiment) showed a stable understanding of numbers (0–10) but still showed significant developmental differences in their ability to place numbers on the number line. The developmental differences between those two groups was expressed by the acquisition of a midpoint as the additional anchor point that helped them more accurately estimate the position of all target numbers on the number line. The aim of this experiment was to validate the finding of the CNP task using the “classic” paper and pencil number line task, similar to the one originally used by Sigler and Opfer [4].

## 3. Experiment 2

### 3.1. Method

#### 3.1.1. Participants

The children in this experiment were recruited from schools that did not participate in the first experiment, but they were from the same socioeconomic class (middle to upper middle class) as the children in Experiment 1. This experiment included a total of 67 right-handed children from second to third grades (31 second graders and 36 third graders).

#### 3.1.2. Materials and Procedure

The number lines were printed on white A4 paper and were the same length as the number line in the digital task. The ends of the number line were marked (0 and 10), and the numbers were presented above the middle of each number line (similarly to the digital task). The target numbers were presented in the following order: 1, 9, 3, 8, 4, 6, 2, 7, and 5. The participant was asked to put a mark on the printed number line corresponding to the presented target number.

### 3.2. Results

We first analyzed the accuracy of the printed NP task (two third graders did not complete the task). An ANOVA with two factors, grade (second, third) × target number (1–9), revealed that the accuracy in second grade was lower (84.8%) than in third grade (87.4%), *F*(1, 65) = 4.4, *p* < 0.05, $\eta_{p}^{2}$ = 0.06. The interaction between grade and accuracy of each target number was significant, *F*(8, 520) = 3.12, *p* < 0.05, $\eta_{p}^{2}$ = 0.04. Planned comparisons examining the grade effect of each target showed there was a significant difference between grades for target numbers 5: *F*(1, 65) = 7.6, *p* < 0.05, $\eta_{p}^{2}$ = 0.01. 

We then analyzed the under- and overestimation of numbers by examining the deviation between the estimated position of each target number and its correct position for each grade (see Figure 8).

Next, we analyzed the differences in the breakpoint between grades (5.86 and 5.54 for second and third grades, respectively). A one-way ANOVA revealed significant differences between grades, *F*(1, 65) = 7.13, *p* < 0.05, $\eta_{p}^{2}$ = 0.09. 

### 3.3. Discussion

The second experiment included the “classic” paper and pencil NP task. The results confirmed the trend observed in the digital task indicating that the differences between the second and third graders were mainly related to the internalization of the midpoint (number 5) as an additional anchor point, facilitating more accurate estimation of the surrounding numbers.

The third experiment we performed examined the correlation between the ability to place each number on the 0–10 number line and the performance of mental arithmetic. Again, the experiment included second and third graders, who on the one hand, demonstrated a solid sense of meaning in the MNL task (consistent with the findings of the first and second experiments), but on the other hand, still differed significantly in their ability to accurately place the midpoint (number 5) on the number line.

## 4. Experiment 3

### 4.1. Method

#### 4.1.1. Participants

A total of 111 right-handed children from the second and third grades were recruited for this study. The second-grade sample consisted of 50 participants (*M* age = 7.46 years, *SD* = 0.3; 27 females). The third-grade sample consisted of 61 participants (*M* age = 8.45 years, *SD* = 0.29; 28 females).

The procedure was approved by the Ministry of Education. Participants were recruited from a local elementary school. The children were predominantly from middle to upper middle class families. Written informed parental consent was obtained for all participants.

#### 4.1.2. Materials and Procedure

All the participants met one-on-one with the experimenter, and the experiments were performed during the school day in a quiet room, in two different sessions. The first session included the same CNP task as that of the first experiment.

The second session included the AF task. In the AF task, the true/false verification task model was used. The task was conducted on a laptop computer with a 17-inch screen, which was viewed from approximately 50 cm. E-Prime software controlled the presentation of stimuli. The experiment started with 10 equation-training trials (that were not analyzed and for which feedback was given). This was followed by 64 one-digit addition and subtraction equations; 32 equations were correct, and 32 equations were incorrect, with four different deviation levels (i.e., ±1, ±3, ±5, and ±7; termed “split”) from the correct value. 

Each trial began with a fixation sign (500 ms), then a black screen (300 ms), and finished with the stimulus, which was presented at the center of the screen and remained visible until the participant responded. The trial finished with another black screen for 1000 ms. Every seven trials, the participants were given a break until they were ready to continue the task (see Figure 9). The “M” and “X” keys of a standard QWERTY keyboard served for correct and incorrect key responses, respectively. The participants were instructed to answer as accurately and as quickly as possible. Split-half reliability was calculated using the Spearman–Brown coefficient [41] and was found to be adequate: *r* = 0.74.

First, the analysis of the CNP task will be presented, followed by the analysis of the AF task. Finally, the correlation analysis between these tasks will be described.

### 4.2. Results

#### 4.2.1. Analysis of the CNP Task

Similar to the first experiment, the analysis only included data acquired for the numbers 1–9. First, the differences in the accuracy of each target number estimation between second and third graders were calculated. The PAE of second graders was 14% and for the third graders was 11%. To test whether this difference was meaningful, we conducted a one-way ANOVA, with PAE as the dependent measure and with grade (second and third) as a between-subject factor and target number (1–9) as a within-subject factor. This analysis revealed a significant main effect of grade, *F*(1, 109) = 19.41, *p* < 0.0001 $\eta_{p}^{2}$ = 0.15; target number, *F*(8, 872) = 49.67, *p* < 0.0001 $\eta_{p}^{2}$ = 0.31; and the interaction grade × number, *F*(8, 872) = 6.60, *p* < 0.0001 $\eta_{p}^{2}$ = 0.06 (see Figure 10).

Next, to characterize the distribution patterns of number estimations on the number line, we computed (for each participant) the *R*^2^ values of the fit for each possible model (the two-linear model, the proportion-estimation model, and the sigmoidal model). We relied on the following sigmoidal equation according to the Levenberg–Marquardt algorithm: $y=\mathrm{Bottom}+\frac{\left(\mathrm{Top}-\mathrm{Bottom}\right)}{\left[1+\exp\frac{\left(\mathrm{breakpoint}-\mathrm{estimated}\;\mathrm{position}\right)}{\mathrm{slope}}\right]}$.

General linear analysis showed that, in both grades, the sigmoidal model with a flexible breakpoint fit the estimation patterns better than the two-linear model and the proportion-estimation model, (see Table 3).

We then examined the breakpoint differences between grades. The results showed that the mean breakpoint of second graders was 5.88, and for third graders it was 5.54. The *t*-test of independent variables between groups showed significant differences, *t*(1, 109) = 3.4, *p* < 0.001, $\eta_{p}^{2}$ = 0.03. Next, we examined the accuracy of the numbers that were used as anchor points (1, 9, and 5). There were no significant differences in the estimation accuracy of number 1: 93.8% and 94.3%, for second and third graders, respectively, *t*(1, 109) = 0.69, *p* = 0.48, $\eta_{p}^{2}$ = 0.006. 

However, we found significant differences between grades in the estimation accuracy of 9 and 5. We found that third graders (vs. second graders) were more accurate in estimating the position of number 5: 82% versus 90%, respectively, *t*(1, 109) = 4.47, *p* < 0.001, $\eta_{p}^{2}$ = 0.04. Third graders (vs. second graders) were also more accurate in estimating the position of number 9: 91% versus 94%, respectively, *t*(1, 109) = 2.2, *p* < 0.05, $\eta_{p}^{2}$ = 0.02. These results confirm our hypothesis that younger children use only the left endpoint as an anchor while elder children use additional anchors, namely, the right endpoint and the midpoint. 

As shown in Figure 11, the second graders tended to underestimate all numbers in contrast to the third graders, who underestimated small numbers and overestimated large numbers. This observation was confirmed by analyzing the estimation bias from the correct position (correct position − estimated position). We conducted a one-way ANOVA with bias as the dependent measure and with grade (second and third) as a between-subject factor and target number (1–9) as a within-subject factor. This analysis revealed a significant main effect of grade, *F*(1, 109) = 8.47, *p* < 0.0005, $\eta_{p}^{2}$ = 0.07; target number, *F*(8, 872) = 54.1, *p* < 0.0001, $\eta_{p}^{2}$ = 0.33; and the interaction grade × number, *F*(8, 872) = 3.71, *p* < 0.0001, $\eta_{p}^{2}$ = 0.03.

We then examined whether the children in each group had more success in placing the numbers closer to the number line ends (1, 9) compared to other number positions (2 and 3 as well as 7 and 8). A quadratic contrast (including the numbers 1–9) conducted for each grade revealed that children in both groups more accurately placed numbers close to the endpoints than other number positions on the number line: in second grade, *F*(1, 49) = 92.12, *p* < 0.001, $\eta_{p}^{2}$ = 0.65, and in third grade, *F*(1, 60) = 58.77, *p* < 0.001, $\eta_{p}^{2}$ = 0.50. Additional analysis showed significant differences between grades, *F*(1, 109) = 6.1, *p* < 0.05, $\eta_{p}^{2}$ = 0.05.

#### 4.2.2. Analysis of the AF Task

First, general accuracy for identifying the correct equations for each grade was high, 93% in second grade and 95% in third grade, and the difference between the grades was insignificant, *F*(1, 109) = 2.4, *p* = 0.12, $\eta_{p}^{2}$ = 0.002. However, the main effect for the split factor (split 0 refers to correct equations; the other splits of 1, 3, 5, and 7 refer to equations with an incorrect value that was larger or smaller than the correct one) was significant, *F*(1, 436) = 8.65, *p* < 0.0001, $\eta_{p}^{2}$ = 0.07. The interaction split × grade was insignificant, *F*(4, 436) = 0.2, *p* = 0.93, $\eta_{p}^{2}$ = 0.001.

Next, we calculated the difference in RT that was required for the identification of correctness/incorrectness at each split condition (see Figure 12). The two-factor between-group ANOVA of grade (second and third) × split (0, 1, 3, 5, and 7) revealed a significant difference between grades, *F*(1, 109) = 9.36 *p* < 0.005, $\eta_{p}^{2}$ = 0.08, and between splits, *F*(4, 436) = 14.12, *p* < 0.0001, $\eta_{p}^{2}$ = 0.11. The interaction split × grade was insignificant, *F*(4, 436) = 1.6, *p* = 0.16, $\eta_{p}^{2}$ = 0.01. 

To confirm our hypothesis according to which children with more mature representation of the number line are expected to quickly identify the correctness of each equation, we performed a correlation analysis. In addition, it was important to figure out whether the obtained correlation was reflected by the use of anchor points.

#### 4.2.3. Correlation Analysis

Regression analysis between the overall RTs of the AF task and the percentage of overall accuracy of the CNP showed significant regression in second grade, *F*(1, 49) = 4.8, *p* < 0.05, *R*^2^ = 0.09, and in third grade, *F*(1, 59) = 4.09, *p* < 0.05, *R*^2^ = 0.06.

We next calculated each child’s correlation between their mean accuracy for each number in the CNP task and their mean RT for each split (0, 1, 3, 5, and 7) in the AF task. For the second graders, individual differences in the RT of the AF task were found to be related to the estimation accuracy of numbers 1 and 8 on the number line, as seen in Table 4. The negative correlations indicate that children who more accurately determined the location of numbers 1 and 8 on the number line were faster at recognizing the correctness of the arithmetic equations. 

For the third graders, the individual differences in the RT of the AF task were found to be only related to the estimation accuracy of the number 5 on the number line (see Table 4). In other words, children who more accurately placed 5 on the number line were faster at recognizing the correctness of the arithmetic equations. Notice, however, that these correlations were not different between grades, when testing the significance using Fisher R-to-z transformations.

### 4.3. Discussion 

In this experiment, we used two tasks: the computerized number-to-position (CNP) task was used to reflect MNL representation; and the arithmetic facts (AF) task (with numbers up to 10)—using the verification task model—was used to show the relationship between MNL maturation and mental arithmetic performance. Both tasks had time restrictions, thus, revealing the intrinsic representations of numbers on the MNL on the one hand, and the ability to retrieve arithmetic facts on the other.

Again, we found that the estimation pattern in the CNP task of both grades was best fit by the sigmoidal function. The results showed that children tended to classify numbers by their value (either small or large). Moreover, our findings indicated that second graders used only the left endpoint (0) as an anchor and treated numbers 1–5 as “small numbers,” yielding an underestimation pattern. In contrast, third graders used additional anchors: the right endpoint (leading to overestimation of large (6–9) numbers) and the midpoint. 

The AF task results showed differences between groups in the speed of the given arithmetic equation. We found that third graders were generally faster in identifying incorrect equations than second graders. These findings were anticipated because third graders had more exposure to arithmetic. Both groups demonstrated the split effect.

Looking at the individual differences between the children of the same grade, we found a relationship between the accuracy in the CNP task and the RTs of the AF task. Second graders showed significant correlation between the accuracy of estimating numbers 1 and 8 in the CNP task and the speed of RT in verifying the accuracy of arithmetic equations (Table 4). The correlations between the accurate estimation of numbers close to the left end (number 1) and the high speed of responses in the AF task supports our hypothesis. 

However, instead of the anticipated correlation with the number close to the right endpoint (number 9), a correlation was found with the number 8. A possible explanation for the above observation could be the fact that the second graders still used only the left endpoint as an anchor, while the response speed in the AF tasks closely correlated with their ability to place the number more accurately by basing their decision solely on the left endpoint anchor. Most importantly, third graders showed a significant correlation between their ability to accurately place the midpoint (number 5) in the CNP task and faster performance in the AF task (see Table 4).

## 5. General Discussion

The main purpose of this study was to examine the developmental patterns of estimating number positions on a 0–10 number line in the first to sixth grades and to demonstrate a correlation between MNL maturation and mental arithmetic performance. We chose to focus on the 0–10 number range in an attempt to avoid developmental differences that could stem from different encoding processes involved for multidigit numbers and for one-digit numbers [10,24,25]. In the current study, we conducted experiments encompassing both computerized (Experiment 1) and “classic” paper and pencil (Experiment 2) methodologies to perform number-to-position (NP) tasks. 

The computerized methodology was implemented on a touchscreen. Utilizing touchscreens for performing the NP task was motivated by two reasons; first, the computerized NP task (CNP) on a touchscreen restricted the response time, which allowed revealing the immediate and internal representation of numbers on the MNL. Second, recent studies [45,46] demonstrated that ballistic tasks—in our case, the NP task—yield better performance when using a touchscreen compared to that observed in “classic” experiments based on paper and pencil methodologies. 

This observation can be explained by the fact that the modern generation of children is growing up in a world saturated with digital technology and media where smartphones and touchscreens are increasingly replacing traditional paper and pencil tools. Thus, the touchscreen tasks enable children to overcome the hand–eye coordination problem and “directly” interact and manipulate objects via their fingers. Nonetheless, we conducted the second number line task experiment by employing the “classic” paper and pencil methodology, similar to that originally used by Siegler and Opfer [4], in order to formally validate the findings of the CNP task.

Our study generated several important findings. First, we found that the sigmoidal function model best fit the pattern of estimated number positions for second through sixth graders. This model incorporates both the proportion judgment model [11] and the two-linear model with a flexible breakpoint [9]. The results of estimated positions of each number revealed that the second through sixth graders showed more accurate placement of numbers close to the endpoints (1, 9), underestimated the position of small numbers (2–5) and overestimated the position of large numbers (7–8). 

However, the first graders showed more accurate placement of number 1 only (close to the left edge) and underestimated the position of numbers 2–9. These results demonstrate the developmental change in the use of endpoints as anchors: first the left endpoint, then the right endpoint. Consistent with the two-linear model with a flexible breakpoint, the examination of breakpoints showed that the breakpoint of first graders was 6, and then in the subsequent grades, the breakpoint linearly decreased until it stabilized at 5.16 in sixth grade. The stabilization of a breakpoint close to 5 indicates the establishment of an additional anchor point, namely, the midpoint, which is crucial for 0–10 number representation [47]. 

The regression analysis between the breakpoint and overall accuracy showed that once the breakpoint decreased to 5, children better estimated the position of all numbers. Such a developmental change in the establishment of anchor points can be an index of the maturation of proportional reasoning [48], resulting in more mature MNL representation [11]. These results are supported by other studies; for example, a study by Huber and colleagues [49] showed an improvement in performance in adults with developmental dyscalculia (DD) once an anchor point on the number line was presented to the participant. 

Reid and colleagues [18] investigated estimation strategies that children used when placing numbers 1–9 on a 0–10 number line. They found that children tended to count from 0 (left anchor) for the first six numbers; in contrast, when estimating the position of numbers 7–9 on a number line, the children often used backward counting, using the right anchor point. 

It should be noted that the same trends were observed in the “classic” NP task, confirming the findings of this study, as well as confirming the statement that the ability to more accurately estimate the position of each number is due to the use of additional anchor points. These observations make us more confident that the results obtained by the computerized task are not due to the motor skills requirements of the touch-screen methodology. 

It is interesting to note that similar strategies based on the proportional reasoning and the gradual acquisition of breakpoints, as we have demonstrated in the NP tasks, have been suggested in additional areas of cognitive developments; for example, in the area of spatial cognition [50,51,52,53,54]. Similarly to what we showed regarding the development of the MNL, it has been found that when mapping distances, children and adults also visualize visible midline boundaries that divide larger spaces in half. In particular, they reduce the size of the space, thereby, facilitating accurate scaling of larger test spaces [55,56].

Next, the AF task results showed differences between groups in the speed required to verify the accuracy of an arithmetic equation. We found that third graders were generally faster in identifying incorrect equations than second graders were. These findings are reasonable and not unexpected because third graders have had more exposure to arithmetic. Both groups showed the split effect.

Looking at the individual differences between children in the same grade, we found a relationship between the accuracy in the CNP task and the RTs of the AF task. Second graders showed significant correlations between the accuracy of estimating numbers 1 and 8 in the CNP task and the speed of RT in verifying the accuracy of arithmetic equations. The correlations between the accurate estimation of numbers close to the left end (number 1) and fast responses in the AF task correspond with our hypothesis. 

However, an additional correlation was expected with the number close to the right endpoint (number 9); instead, a correlation was found with the number 8. Perhaps these results show that second graders struggle more with the position of 8 because they still use only the left endpoint as an anchor, and once they are able to place this number more accurately, it correlates with speed in the AF tasks.

Most importantly, with third graders, there was a significant correlation between the ability to accurately place the midpoint (number 5) in the CNP task and faster performance in the AF task. This is consistent with the idea that the establishment of 5 as a midpoint anchor in the 0–10 number line reflects the development of more mature MNL representation, which could explain the faster performance in the arithmetic fact retrieval task. 

Still, the present design, which is correlational in nature, does not enable such a causal conclusion. At this point, we cannot determine whether the faster RTs in determining correct and incorrect arithmetic equations were indeed affected by MNL development, or rather, were a consequence of the development of other factors, such as practice and exposure to the numbers 0–10 (e.g., [5,9]), which might impact arithmetic performance. Further research would be needed to explore this idea.

To conclude, the present study presents an alternative model to represent the developmental patterns in estimating number position along the 0–10 number line. We propose the sigmoidal curve as the best fit to represent the estimated positions of numbers 0–10. This fit indicates the differences between the breakpoints, which reflect whether the children use the midpoint as an additional anchor point along with the use of endpoints. Examining the estimated position of numbers of first through sixth graders, we found two developmental leaps: one that occurred between the first and second grades; this leap is characterized by the use of the right endpoint as an additional anchor point. 

The other leap occurred between second and third grades when children began to use the midpoint as a third anchor point. Our results showed that using the midpoint as an anchor is critical for more accurate estimation of all numbers on the number line and in the performance of the mental arithmetic task both indicates more mature representation of the MNL.

The changes in the breakpoints that divide the number line into two segments may have been affected by the association between fingers and numbers and can be explained by the sub-base 5 finger theory [57]. The sub-base 5 finger theory claims that children use fingers to represent numbers up to 10; to represent numbers 1–5, they usually count the fingers on one hand first, and to represent numbers 6–10, they count on the other hand [58,59,60]. Our results, which are consistent with this theory, show that children who did not use the number 5 as an anchor point, compress the position of numbers 1–5 that represent one hand and relatively compress the position of numbers 6–9 that represent the other hand, which is crucial for many arithmetic competences (e.g., [61,62]).

To avoid any differences in the encoding process of multiple digit numbers, our study used a narrow range of numbers. Future studies should test number lines with a wider range of numbers or with different endpoints (e.g., 3 and 23). This would enable a test of whether the proposed model providing the best fit for estimated number position in the 0–10 range would also provide the best fit for other ranges of the number line. Moreover, future studies should focus on the examination of how children divide different ranges and the way they use the anchor points. Understanding the developmental process of employing the anchor points will contribute to the progress in promoting maturation of MNL representation.

### Implication for Elementary School Math Teachers

In recent years, there has been growing interest in the number line model to help children in their development of a number sense and mathematical concepts. In particular, the emphasis was set on helping the children to develop greater flexibility in mental arithmetic by focusing on active construction of mathematical meaning, on the number sense, and on the understanding of number relationships.

Most often, the 0–10 number line is displayed above the whiteboard right above the alphabet and is used to help young children to memorize and to practice counting with ordinal numbers. The current study sheds light on the development process in utilizing the number lime model. We suggest that the observed division of a number line into two segments may have been affected by the finger counting and “ten-frame” models, to visualize the concrete number of objects up to 10. Both models are based on the sub-base 5 [63,64,65], according to which one hand or a half frame represents numbers in the range of 1–5 and other hand or the other half frame represents numbers in the range of 6–10. 

Using of the above sub-base is typical of first and second graders and is consistent with the results obtained in the current study indicating that first and second graders make the breakpoints close to number 6. This means that children mentally create two groups of numbers lying in the ranges of 1–5 and 6–9, respectively. Such a division apparently occurs as a result of transferring the concrete visualization of numbers from the finger-counting and the ten-frame models to the number-line model. Third graders, however, begin to move away from the use of concrete models, and instead use the number line as a tool for better understating of mental arithmetic, number sense, and number relationships. 

Moreover, the currently presented results showed that a more mature representation of a MNL, which is characterized by the consistent use of anchor points (first the endpoints and then the midpoint), related to faster performance in the AF retrieval task. Unfortunately, at this point, we cannot distinguish whether the faster RTs in determining correct and incorrect arithmetic equations are indeed affected by MNL development or, rather, are a consequence of the development of other factors, such as practice and exposure to the numbers 0–10 (e.g., [5,9]). Further research would be needed to explore this correlation. An additional promising direction for future studies could be to look for correlations between individual differences in mathematical performance and in the CNP tasks, taking into consideration individual differences in motor skills. This issue is a limitation and was beyond the scope of the present study.

## Figures and Tables

**Figure 1 brainsci-12-00335-f001:**
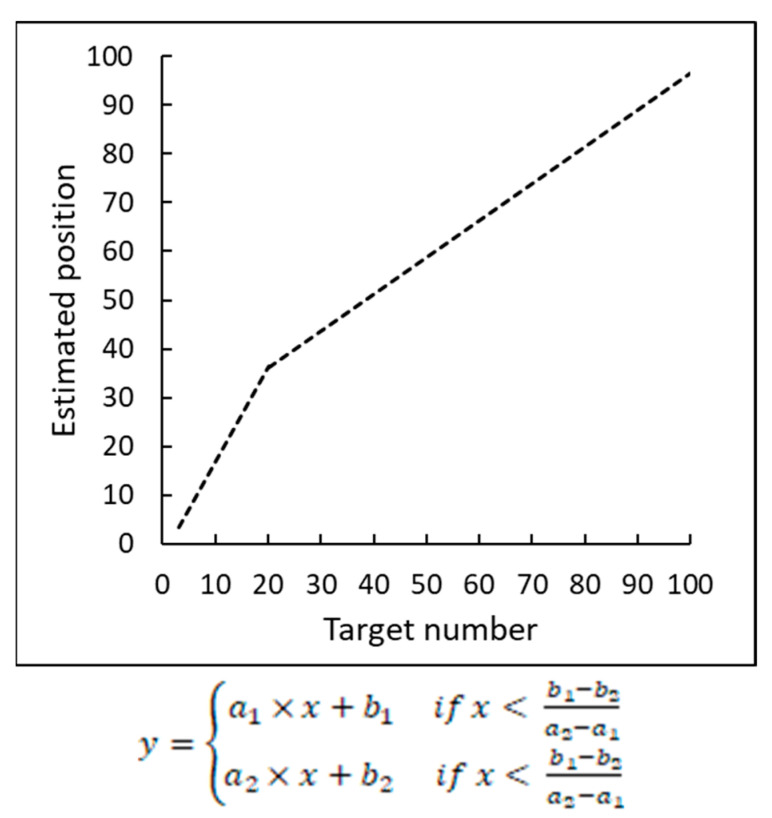
A representative schematic of a curve fitting the two-linear model of Ebersbach et al. [9] with a flexible change point equal to 20. The values of xϵ(0, 20) correspond to the range of more familiar numbers and are fit by the line of the steeper slope while the values of xϵ(21, 100) correspond to the range of less familiar numbers and are fit by the line of the less steep slope.

**Figure 2 brainsci-12-00335-f002:**
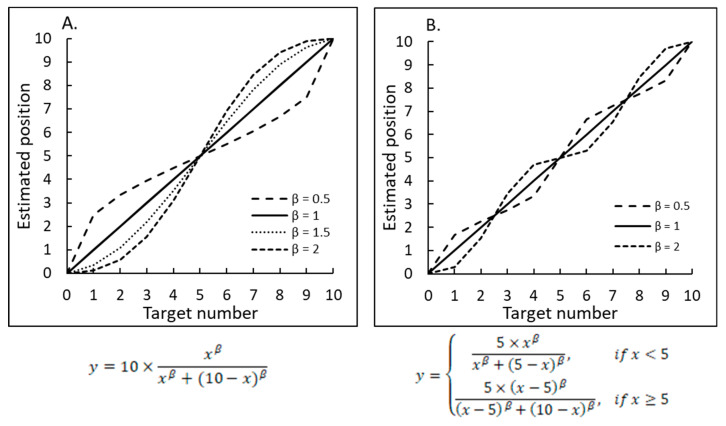
A representative schematic of curves fitting the estimated number position according to the proportion judgment model [12,13]. The shape of the curve depends on the value of *β*. (**A**) The curve fits the estimated number position when using numbers on the edges as anchors. (**B**) The curve fits the estimated number position when using the midpoint as an additional anchor point.

**Figure 3 brainsci-12-00335-f003:**
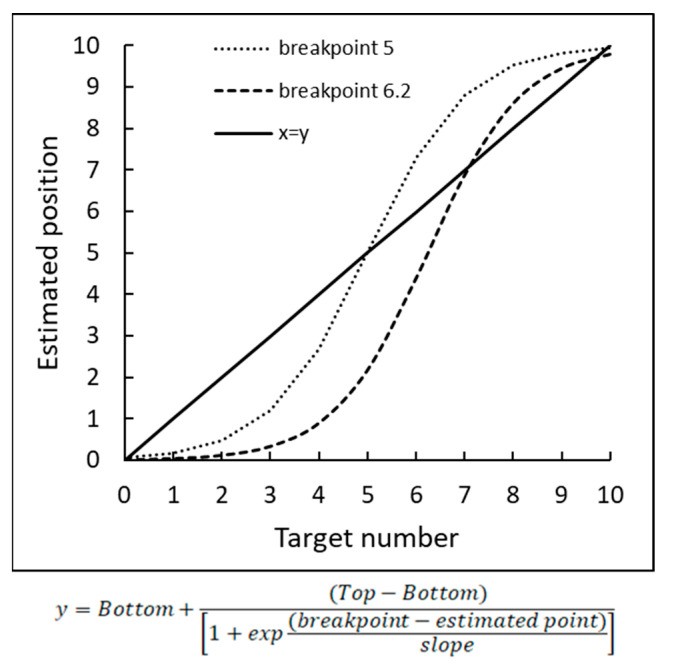
Illustration of number distribution according to the sigmoidal curve function.

**Figure 4 brainsci-12-00335-f004:**
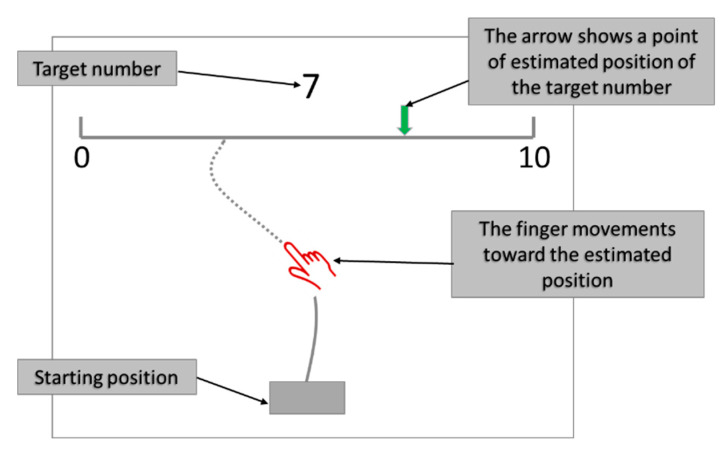
A schematic example of the task with a participant’s finger movement.

**Figure 5 brainsci-12-00335-f005:**
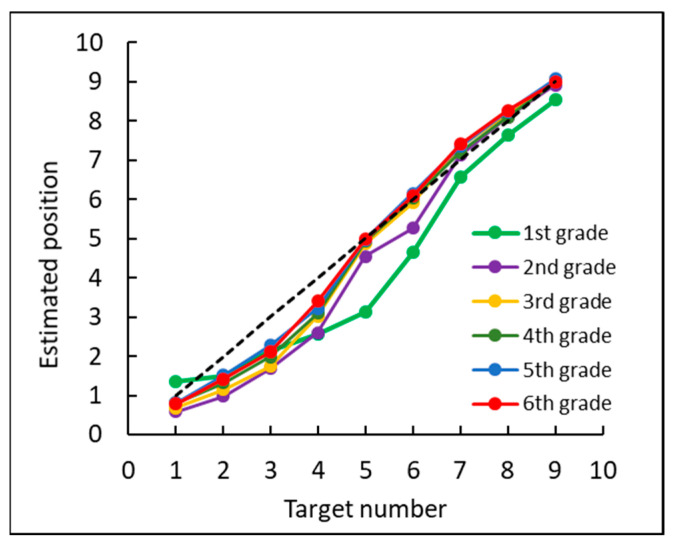
The mean estimated position of each target number for each grade. The dashed black line indicates the correct position of each target number (*x* = *y*).

**Figure 6 brainsci-12-00335-f006:**
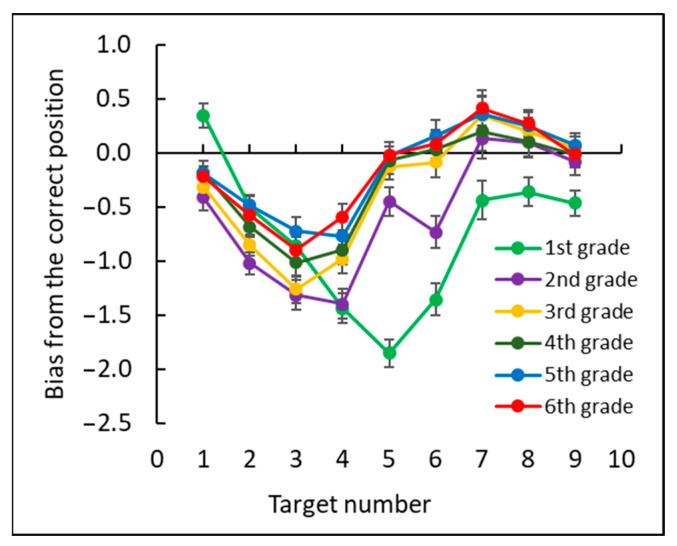
The mean bias value: the deviation between the estimated position of each target number and its correct position for each grade.

**Figure 7 brainsci-12-00335-f007:**
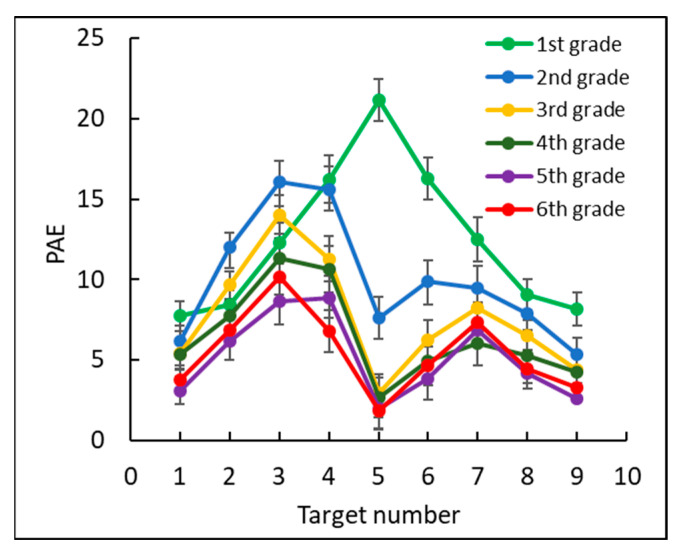
The percentage of absolute error (PAE) by target number for each grade.

**Figure 8 brainsci-12-00335-f008:**
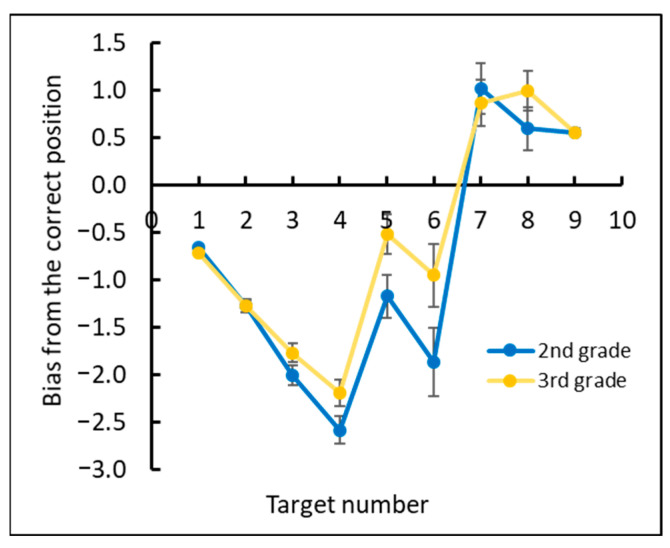
The mean bias value: the deviation between the estimated position of each target number and its correct position for each grade.

**Figure 9 brainsci-12-00335-f009:**
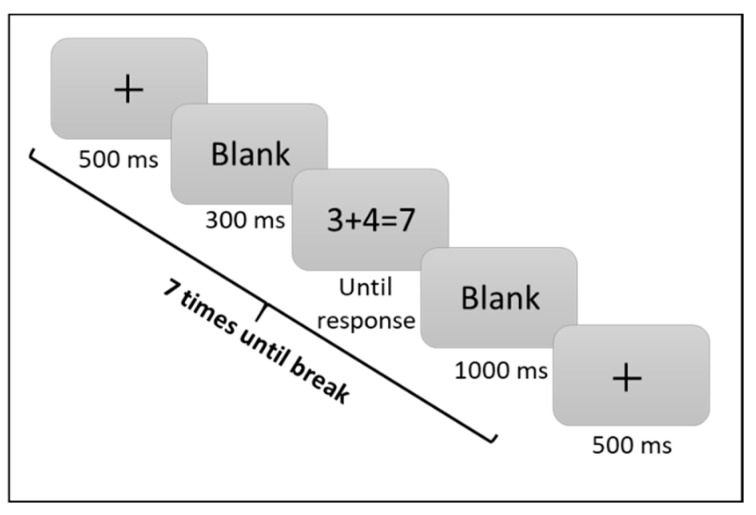
The scheme of the AF task according to the true/false verification task model.

**Figure 10 brainsci-12-00335-f010:**
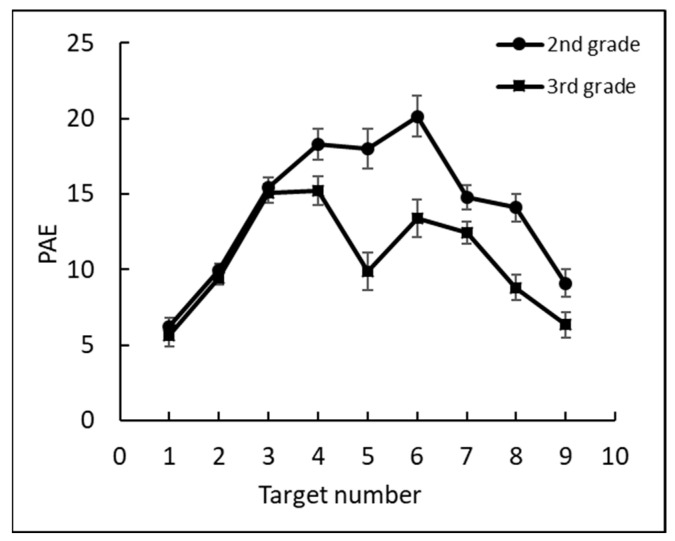
The percentage of absolute error (PAE) by target number per grade.

**Figure 11 brainsci-12-00335-f011:**
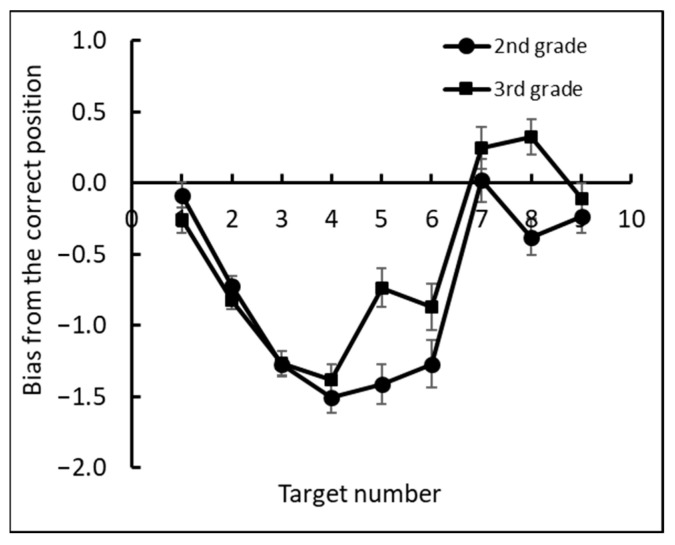
The mean bias value: the deviation between the estimated position of each target number and its correct position for each grade.

**Figure 12 brainsci-12-00335-f012:**
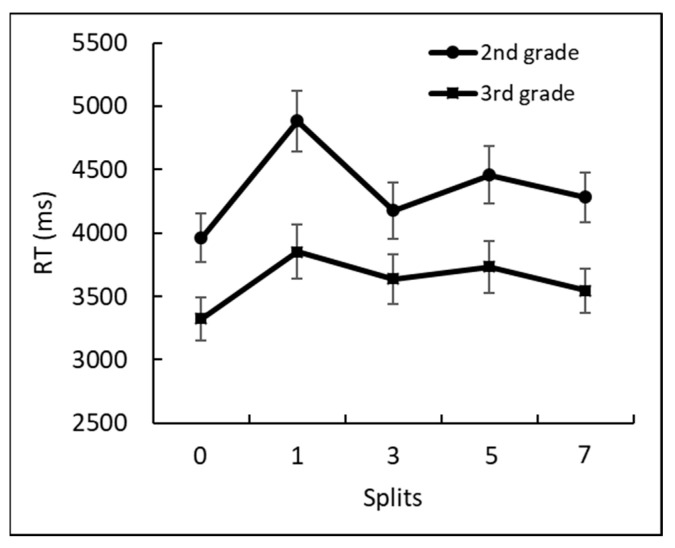
The reaction times for each split (i.e., deviation level from the correct value) by grade.

**Table 1 brainsci-12-00335-t001:** Group, sample size, mean age, and gender distribution.

Group	N	The Mean Age in Years (SD)	Males	Females
1st grade	28	6.9 (0.4)	16	12
2nd grade	27	7.8 (0.3)	15	12
3rd grade	31	8.9 (0.5)	14	17
4th grade	29	9.8 (0.3)	12	17
5th grade	31	11.0 (0.3)	10	21
6th grade	35	11.9 (0.4)	21	14

**Table 2 brainsci-12-00335-t002:** Statistics of the various models by age group in Experiment 1.

Age Group	Model	*R* ^2^	Model Comparison	Repeated Contrasts				AIC
				*F*	*df*	*p*	$\eta_{p}^{2}$	
1st grade	2L	0.85	2L vs. P	12.13 **	1, 27	<0.001	0.31	25%
	P	0.7	2L vs. S	2.97		=0.09	0.10	7%
	S	0.88						68%
2nd grade	2L	0.93	2L vs. P	4.11 *	1, 26	=0.053	0.14	11%
	P	0.89	2L vs. S	20.24 **		<0.001	0.44	4%
	S	0.96						85%
3rd grade	2L	0.92	2L vs. P	1.46	1, 30	=0.24	0.05	--
	P	0.94	P vs. S	15.13 **		<0.001	0.34	6%
	S	0.98						94%
4th grade	2L	0.95	2L vs. P	0.11	1, 28	=0.74	0.00	7%
	P	0.95	P vs. S	23.31 **		<0.001	0.45	7%
	S	0.97						86%
5th grade	2L	0.95	2L vs. P	0.43	1, 30	=0.52	0.01	13%
	P	0.98	P vs. S	13.49 **		<0.001	0.31	3%
	S	0.99						84%
6th grade	2L	0.97	2L vs. P	0.001	1, 34	=0.97	0.00	9%
	P	0.97	P vs. S	8.85 *		<0.01	0.21	14%
	S	0.99						77%

2L—Two-Linear, P—Proportion, S—Sigmoidal. * *p* < 0.05 and ** *p* < 0.01.

**Table 3 brainsci-12-00335-t003:** Statistics of the various models by age group in Experiment 3.

Age Group	Model	*R* ^2^	Model Comparison	Repeated Contrasts			
				*F*	*df*	*p*	*ƞ* ^2^
2nd grade	2L	0.883	2L vs. P	38.58	1, 49	<0.001	0.44
	P	0.767	2L vs. S	63.07		<0.001	0.56
	S	0.927					
3rd grade	2L	0.936	2L vs. P	7.89	1, 60	<0.001	0.31
	P	0.888	2L vs. S	18.8		<0.001	0.66
	S	0.998					

2L—Two-Linear, P—Proportion, S—Sigmoidal.

**Table 4 brainsci-12-00335-t004:** Correlation between CNP task accuracy and AF task RT in Experiment 3.

	Each Number of the CNP Task								
	1	2	3	4	5	6	7	8	9
2nd grade									
Split 0	−0.326 **	−0.128	−0.022	−0.256 *	−0.153	−0.004	−0.096	−0.440 **	−0.057
Split 1	-0.243 *	−0.015	0.009	−0.080	−0.129	−0.057	−0.203	−0.387 **	−0.073
Split 3	−0.308 *	−0.014	0.051	−0.132	−0.137	.023	−0.098	−0.272 *	−0.112
Split 5	−0.380 **	0.096	0.037	−0.248 *	−0.232	−0.038	−0.042	−0.283 *	−0.063
Split 7	−0.355 **	−0.078	−0.005	−0.225	−0.212	−0.053	−0.129	−0.274 *	0.093
3rd grade									
Split 0	−0.133	−0.178	−0.144	−0.026	−0.326 **	−0.151	−0.103	−0.178	−0.072
Split 1	−0.072	−0.137	−0.130	−0.001	−0.279 *	−0.191	−0.041	−0.192	−0.061
Split 3	0.010	−0.131	.040	−0.075	−0.288 *	−0.184	0.041	0.046	−0.064
Split 5	−0.045	−0.188	−0.095	−0.053	−0.337 **	−0.217 *	−0.050	−0.206	0.033
Split 7	−0.059	−0.131	−0.085	0.024	−0.249 *	−0.135	−0.068	−0.205	0.039

* *p* < 0.05 and ** *p* < 0.01.

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
