# Peer review of "Development of the Mental Number Line Representation of Numbers 0–10 and Its Relationship to Mental Arithmetic"

_brainsci, 2022, doi:10.3390/brainsci12030335_

Round 1
Reviewer 1 Report
I thought this was a very interesting and important paper. The issue of how the number line is represented mentally, and how this representation changes with development, is a fundamentally important one, both for developmental psychology and for education. The authors point out that the relation between numeric, integer value and children's estimates cannot, or should not be represented as either a linear or logarithmic function (as has been debated in prior research), but rather by a sigmoid function. The "breakpoint" of this function seems to be what changes with development.
The paper is very well written and clear, The methodological explication is adequate. I also very much appreciate the consideration of the implications of the work for education (although obviously this is not an easy question, and much more research is required).
Suggestion: In finding that the critical aspect is the "breakpoint" of the sigmoid function, the authors may want to make contact with other literatures that have considered the development of such "breakpoints" in other cognitive domains. I'm thinking here of research on the development of spatial cognition and the development of spatial categories, as demonstrated by Huttenlocher, Newcombe, Plumert, Hund, and several other authors. I'm not sure whether the similarity here is meaningful or shallow, but it might be worth considering.
The tables are a little hard to read. There is multiple repetition of labels at different levels of the tables. I suggest that authors put the tables into APA style, if that is acceptable to this journal.
Overall, an interesting paper with important findings and implications.
Author Response
Dear Reviewer,
We are thankful to the reviewer for recognizing the potential contribution of our study to the community and for his/her suggestions. Our replies to specific comments of the reviewer are as follows:
Comment:
In finding that the critical aspect is the "breakpoint" of the sigmoid function, the authors may want to make contact with other literatures that have considered the development of such "breakpoints" in other cognitive domains. I'm thinking here of research on the development of spatial cognition and the development of spatial categories, as demonstrated by Huttenlocher, Newcombe, Plumert, Hund, and several other authors. I'm not sure whether the similarity here is meaningful or shallow, but it might be worth considering.
Response:
We agree with reviewer's comment and have added a paragraph discussing the similarities between studies on the development of spatial cognition and the development of spatial categories and our study. The references to the relevant literature suggested by the reviewer were added as well. The paragraph was added in the General Discussion section and appears on lines 590-596 of the revised version of the paper.
Comment:
The tables are a little hard to read. There is multiple repetition of labels at different levels of the tables. I suggest that authors put the tables into APA style, if that is acceptable to this journal.
Response:
Thank you for the suggestions, we edited all the tables and tried to make them more readable.
Reviewer 2 Report
This is a comprehensive and well-executed study. The results are interesting and well presented. I have only a few considerations that I would like to advance.
From a general point of view (but also in terms of clinical application) how do the authors think this test may be affected by the development of motor skills (e.g. tracking accuracy)? The literature reports that mathematical and motor skills can be correlated; could this link represent a limitation for the application of this test? (compared to a classic number line test without motor tracking and without time pressure)
I would point out that this question could also partially be addressed by the data collected in the current study by 1) looking at the control experiment in which the authors measured tracking accuracy without numbers presentation and 2) looking at the "goodness" the motor trajectories measured in the main experiment(s). Is there a correlation between motor trajectories precision/accuracy (or other parameters) and number line scores? Does the correlation between mathematical performance and number line found here remain significant after controlling for motor skills parameters as a covariate?
Author Response
Dear Reviewer,
We are thankful to the reviewer for recognizing the submitted paper as a "comprehensive and well-executed study". Our replies to specific comments of the reviewer are as follows:
Comment:
From a general point of view (but also in terms of clinical application) how do the authors think this test may be affected by the development of motor skills (e.g. tracking accuracy)? The literature reports that mathematical and motor skills can be correlated; could this link represent a limitation for the application of this test? (compared to a classic number line test without motor tracking and without time pressure)
Response:
We are thankful to the reviewer for this comment. We are aware of the plausible influence of motor skills on the results of the CNP task. In the revised version of the paper, we have now stressed that all the participants successfully completed two trainings, see lines 201-203.
Regarding the reviewer's comment about plausible correlations between the mathematical and motor skills, which could be a limitation of the CNP test, we have added a note about the same trends seen in "classical" and related computerized tasks. The note appears on lines 584-589 of the revised version of the paper.
Comment:
I would point out that this question could also partially be addressed by the data collected in the current study by 1) looking at the control experiment in which the authors measured tracking accuracy without numbers presentation and 2) looking at the "goodness" the motor trajectories measured in the main experiment(s). Is there a correlation between motor trajectories precision/accuracy (or other parameters) and number line scores? Does the correlation between mathematical performance and number line found here remain significant after controlling for motor skills parameters as a covariate?
Response:
Both questions raised by the reviewer are indeed important and timely, a proper addressing of which can be the subject of a standalone study. In the current manuscript, however, the focus was on the correlation between the ability to place individual numbers on a number line and performance of the mental arithmetic task. Both of these objectives determined the kind of data acquired in our computerized tasks, which is insufficient to address the questions raised by the reviewer. We refer to these open questions in a concluding statement regarding future required studies on lines 685-688 in the revised version of the paper.
Round 2
Reviewer 2 Report
I have no further comments and I think the paper is now ready for publication.